# On the Use of Styrene-Based Nanoparticles to Mitigate the Effect of Montmorillonite in Copper Sulfide Recovery by Flotation

**DOI:** 10.3390/polym16121682

**Published:** 2024-06-13

**Authors:** Darwin Estrada, Romina Murga, Olga Rubilar, John Amalraj, Leopoldo Gutierrez, Lina Uribe

**Affiliations:** 1Water Research Center for Agriculture and Mining (CRHIAM), Universidad de Concepción, Concepción 4030000, Chile; destrada@udec.cl (D.E.); olga.rubilar@ufrontera.cl (O.R.); lgutierrezb@udec.cl (L.G.); 2Engineering Systems Doctoral Program, Universidad de Talca, Curicó 3340000, Chile; romina.murga@utalca.cl; 3Department of Chemical Engineering, Universidad de la Frontera, Temuco 4780000, Chile; 4Institute of Natural Resources Chemistry, Universidad de Talca, Talca 3480094, Chile; jamalraj@utalca.cl; 5Department of Metallurgical Engineering, Universidad de Concepción, Concepción 4030000, Chile; 6Department of Mining Engineering, Universidad de Talca, Curicó 3340000, Chile

**Keywords:** collector, copper sulfides, flotation, montmorillonite, nanoparticles, polystyrene

## Abstract

Clay minerals have different negative effects on the froth flotation process such as low adsorption of collectors on valuable minerals, increased pulp viscosity, and the reduction in recovery and grade concentrates of copper sulfides. This study aims to evaluate the use of polystyrene-based nanoparticles (NPs) for the froth flotation of chalcopyrite and their ability to mitigate the negative effect of montmorillonite on the recovery of this sulfide. The experimental stage consisted of preparing a type of polystyrene-based nanoparticle (St-CTAB-VI), which was analyzed by dynamic night scattering (DLS) to establish its hydrodynamic size. Then, the effect of NPs on chalcopyrite’s angle’s in the presence and absence of montmorillonite (15%) was evaluated and compared with the contact angle achieved using potassium amyl xanthate (PAX) and a mixture of PAX and NPs. In addition, zeta potential measurements were carried out to investigate the interactions between the chalcopyrite and the montmorillonite or the NPs under fixed concentrations and microflotation tests were performed employing different times to evaluate the chalcopyrite recovery in the presence of montmorillonite, using NPs and mixtures with PAX. Finally, turbidity analysis as a function of time was performed to evaluate the occurrence of sedimentation and flocculation phenomena in suspensions of 15% montmorillonite in the presence and absence of chalcopyrite, nanoparticles, and mixtures of NPs and PAX. The results indicated that the mixture of NPs and PAX contributed to increasing the contact angle of chalcopyrite in the presence of montmorillonite. This can be associated with the presence of molecular and nanometric collectors that generated a higher hydrophobicity on the chalcopyrite particles, contributing to reducing the presence of clay minerals on the mineral surface. In addition, the mixture of NPs and PAX promoted the generation of nanoparticles on the sulfide mineral surface, which helps to detach the slime and facilitate the bubble/mineral attachment step during flotation.

## 1. Introduction

The aging of major copper deposits has caused a continuous reduction in mineral resources and one of the main challenges today in the field of mineral processing worldwide is the need to process ores containing high levels of clay minerals [1,2,3,4,5,6,7,8,9,10,11,12,13]. Clay minerals refer to a group of hydrous aluminum phyllosilicates that have an anisotropic structure and usually encompass colloidal sizes. They are fundamentally built of tetrahedral silica (T) and octahedral (O) layers, which join with specific proportions, 1:1 (TO) and 2:1 (TOT), presenting two surfaces that are crystallographically different: the faces, which tend to show anionic charge, and the edges, that present anionic or cationic charge depending on the pH [14].

Kaolinite and smectite are the most common clay minerals, which are often associated with copper, gold, and other valuable minerals [15]. Kaolinite is a non-swelling clay with a 1:1 alumina-to-silica layered structure. It has low chemical reactivity [16] and the rheological behavior is related to the grade of crystallinity [7]. Du et al. [17] noted that poorly crystallized kaolinite yielded slower settling rates and lower settled bed density [17]. In addition, montmorillonite belongs to the smectite group, with a 2:1 alumina-to-silica layered structure. It is a swelling clay which can take up to 10 times its weight and increases its volume by 20 times and it is a major component of bentonite [18]. Bentonite slurries display a significant yield stress even at low concentrations due to the high swelling and flocculation of fine clay particles producing a viscous gel-like structure [19]. Farrokhpay and Ndlovu found that montmorillonite is the clay mineral that has the strongest negative effect on chalcopyrite recovery, most probably because of the significant differences in crystallinity, cation exchange capacity, and swelling degree [6].

Some mechanisms have been suggested to explain the adverse effect of the presence of clay minerals on the process of flotation of copper sulfide ores, i.e., slime coatings, coating of bubbles with clay particles, high reagent consumption (reagents uptake), and changes in froth stability and pulp rheology. According to the literature, the phenomenon referred to as slime coating has been proposed to be one of the main mechanisms that negatively affects the efficiency of froth flotation [8,9,10,20,21]. Slime coating consists in the deposition of colloidal clay particles on the surface of larger particles of valuable minerals, which induces hydrophilicity, and therefore, the efficiency of the flotation process is reduced [22].

To remove the clay slime coating from valuable mineral surfaces, both physical and chemical pretreatments have been developed. Physical treatment includes the application of high-intensity agitation, desliming hydrocyclone, and ultrasonic treatment to detach the clay minerals [23,24]. On the other hand, in relation to chemical treatment there are dispersants which can adsorb onto the surface of the particles, creating electrostatic and/or steric repulsions that stabilize the suspension [1,2,25,26,27,28,29,30]. These reagents usually correspond to polysaccharides [28]. Other reagents that have been used as dispersants are lignosulfonates. Liu and Peng demonstrated that lignosulfonates improved the flotation of a problematic coal containing clay minerals [4,25,29]. Ramirez et al. found that dispersants such as sodium hexametaphosphate (SHMP) and sodium silicate (SS) can restore chalcopyrite flotation in the presence of kaolinite in seawater [1].

Recently, nano/collectors have attracted a lot of interest because they have the potential to give much more efficient attachment to air bubbles than molecular collectors [31,32]. In relation to the behavior of nanoparticles, it has been demonstrated that polystyrene nanoparticles (NPs) functionalized by the imidazole group (St-CTAB-VI) as collectors resulted in higher recoveries of sulfide minerals like pentlandite [13] and chalcopyrite [33]. In relation to the presence of clay minerals and the slime coating phenomenon it has been reported that large nanoparticles adsorbed on pentlandite surfaces may be less susceptible to the slime coating produced by serpentine than conventional collectors (a kaolin group clay mineral) [13]. Contact angle and zeta potential measurements, microflotation tests and turbidity analysis, in the presence and absence of montmorillonite, different concentrations of nanoparticles and mixtures with conventional collectors were considered to evaluate the ability of polystyrene nanoparticles (NPs) functionalized by the imidazole group (St-CTAB-VI) to mitigate the negative effect of montmorillonite (smectite group) on the recovery of chalcopyrite by flotation.

## 2. Materials and Methods

### 2.1. Mineral Samples

The mineral samples were supplied by Ward’s Science. After the preliminary crushing, the ores were manually picked to remover impurities, and then, the sample was dry-ground in a planetary mono mill, Pulverisette 6 (Fritsch, Idar-Oberstein, Germany), for 20 min at a speed of 270 rpm to obtain a chalcopyrite sample with a particle size in the range −150/+75 μm for microflotation assays and contact angle measurements and −75 μm for zeta potential measurements. The particle size distribution of the chalcopyrite and volume fraction data are shown in Figure 1 and Table 1. On the other hand, montmorillonite was classified in a lower size of 38 μm using a 400# sieve (Tyler series).

Mineralogical composition analyses were performed using the Miniflex 600 X-ray diffraction (Rigaku, Tokyo, Japan). The identification of the crystalline phases for the chalcopyrite and montmorillonite samples are shown in Figure 2a,b, respectively. The Rietveld least squares method was used for the refinement of crystalline structures and the semi-quantification of phases for both samples, the results, shown in Table 2, indicate that the chalcopyrite sample was composed of chalcopyrite (93.0%), quartz (1.0%), zeolite (4.0%), and digenite (2.0%), while the montmorillonite had 81% montmorillonite, 10% feldspar, 6% quartz, and 3% illite.

### 2.2. Flotation Reagents

The reagents used in this study included Aerofroth 70 as a foaming agent, supplied by Solvay, and potassium amyl xanthate (PAX), supplied by ORICA (Santiago, Chile), as a collector agent. The pH modifiers used were 37% hydrochloric acid, 97% sodium hydroxide granules, and 99.99% sodium chloride, supplied by MERCK (Darmstadt, Germany). The concentration of potassium amyl xanthate (PAX) used for the different tests was 5.6 mg PAX/g chalcopyrite. It is also important to note that all experiments were carried out with 5 mM NaCl solutions prepared with type II water (1 MW/cm^2^).

The preparation of the polystyrene-based organic St-CTAB-VI NPs used the following chemical reagents supplied by Sigma-Aldrich (St. Louis, MO, USA): styrene as monomer, VI (vinylimidazole) as binder, CTAB (cetyltrimethylammonium bromide) as cationic surfactant, and V50 as starter. The manufacturing process of these nanoparticles involves a series of steps, described by Murga et al. (2022) [33]. The nanoparticles were characterized at pH 8 using dynamic light scattering (DLS) and electrophoretic light scattering (ELS) techniques to determine their hydrodynamic radius, electrophoretic mobility, and surface charge.

The results of the NP characterization are presented in Table 3. It can be observed that the synthesized nanoparticles had a hydrodynamic diameter of 75.81 nm, a heterogeneous distribution of nanoparticles according to the obtained polydispersity index (PI) of 20.89% (values higher than 10% indicate high polydispersity), a positive surface charge 50.00 ± 1.26 mV, and an electrophoretic mobility of 3.90 μm·cm/Vs.

### 2.3. Conditioning Stage

Figure 3 shows the conditioning procedure used for each of the measurements made in this study. It is important to note that each measurement was carried out in triplicate to determine the mean value and the standard deviation.

The procedure of addition of reagents and minerals in this stage was the following: the first 2 min corresponded to a pH adjustment of the solution used to pH 8, then the mineral chalcopyrite was added and the suspension was conditioned until 5 min. Next, the selected collector was added and conditioned for 5 min. Finally, the MIBC was added 1 min before finishing the conditioning stage (only in microflotation tests). It is important to note that the montmorillonite (only when it was considered) was added after the conditioning time of chalcopyrite and the stirring was held for 2 min. In addition, when the effect of mixtures of NPs and PAX was studied, the collector PAX was added at the conditioning time with the NPs, and the suspension was stirred for 5 min.

### 2.4. Contact Angle Measurements

Contact angle measurements were made to evaluate the effect of NPs on chalcopyrite’s angle’s in the presence and absence of montmorillonite (15%). The contact angle measurement was carried out using a Lauda 50 surface analyzer (Lauda Scientific GmbH, Lauda-Königshofen, Germany) and employing the sessile drop and Young–Laplace methods to calculate the angle of compressed discs. After the conditioning stage, the compressed discs were prepared using 3 g of chalcopyrite and 10 tonnes of pressure through an industrial press to obtain a surface capable of retaining a drop of water for a certain period. Contact angle measurements were taken at points on the surface of the processed pellet to obtain an average of the measurements. The conditions studied were the following: chalcopyrite in the presence and absence of montmorillonite (15%) at different doses of NPs and NPs + PAX mixtures using fixed doses of NPs.

### 2.5. Zeta Potential Measurements

Zeta potential measurements were carried out using a dynamic light scattering instrument (Litesizer 500; Anton Paar, Ostfildern, Germany) to investigate the interactions between the chalcopyrite and the montmorillonite or the NPs. Zeta potential analyses were made as a function of pH, considering the following samples: chalcopyrite, montmorillonite, chalcopyrite + NPs, chalcopyrite + montmorillonite.

The samples were obtained by mixing 0.714 g of chalcopyrite (<75 µm) in 50 mL of 5 mM NaCl solution, following the same procedure indicated in Figure 3 and considering a concentration of 0.25 g/L of nanoparticles, 14.28 g/L of chalcopyrite, and the presence of 15% of montmorillonite in the suspension. After the conditioning step, 0.4 mL of this suspension was transferred to an Omega cuvette for zeta potential measurements.

### 2.6. Microflotation Tests

Microflotation tests were performed in a 140 mL Partridge–Smith cell, using a nitrogen flow of 0.09 cm^3^/s and a magnetic stirring of 900 rpm. These tests were performed to evaluate the chalcopyrite recovery in the presence of NPs and mixtures with PAX when montmorillonite is present in the suspension. The microflotation tests considering the same conditioning stage are shown in Figure 3, evaluating different conditioning times (between 5 and 12 min) and considering a fixed concentration of NPs, which was established according to the results obtained in previous tests: 5.6 mg/g PAX and 15 ppm MIBC. The flotation stage was performed for 2 min with manual scraping every 10 s. Concentrates and tailings obtained in the tests were first sieved on the 200 mesh to eliminate the presence of montmorillonite and afterwards filtered and dried. Finally, the recovery was calculated considering the quantities between the concentrate mass and the feed mass.

### 2.7. Turbidity Tests

Turbidity measurements were carried out using a stability analyzer instrument (Turbiscan Classic 2; Formulaction, Toulouse, France), an instrument specialized in the analysis of destabilization mechanisms. The measurements are based on a change in average size (merge), which results in a change in backscattering (BS) and transmission (T) signals as a function of time. The purpose of using this equipment was to analyze the effect of the presence of chalcopyrite, NPs, and mixtures of NPs + PAX in the occurrence of sedimentation, flocculation, or coalescence phenomena in suspensions of 15% montmorillonite. It is important to note that the fixed concentrations of NPs and PAX were established according to the results obtained in prior tests.

## 3. Results and Discussion

### 3.1. Contact Angle

Figure 4a shows the variation in the contact angle in chalcopyrite samples in the presence and absence of montmorillonite (15%) using different concentrations of polymeric NPs as the collector.

In the figure, it can be observed that the contact angle of chalcopyrite decreases by approximately 10 degrees in the presence of montmorillonite. On the other hand, it was also observed that the presence of St-CTAB-VI NPs increased the contact angle of chalcopyrite, even at the lowest addition of NPs, obtaining a maximum contact angle of chalcopyrite of 89 degrees in the absence of montmorillonite and 77 degrees in the presence of this type of clay mineral. These values were slightly higher than those obtained in the presence of 2.8 mg/g of PAX, in which the measured contact angles were 87 degrees and 68 degrees, respectively. According to these results, it is possible to note that the presence of montmorillonite has a similar effect to the NPs when a conventional collector is used. It is well known that anisotropic charges on their basal surfaces and edges cause clay particles to coat the surface of valuable minerals by electrostatic attraction. As a result, it decreases the surface hydrophobicity, which is evidenced by the reduction in contact angle values. Figure 4b shows the results of comparative contact angle measurements of chalcopyrite obtained at pH 8 in the presence of montmorillonite (15%), considering separately the addition of 5.6 mg/g of PAX, 7.5 mg/g of St-CTAB-VI NPs, and a mixture of NPs + PAX. From the figure, it can be observed that the contact angles obtained in the presence of nanoparticles and PAX were alike, but the joint action of NPs and PAX achieved a significant increase in the contact angle of chalcopyrite. A first adsorption of the PAX collector may be contributing to a greater interaction of the NPs with the mineral surface and reducing the interaction with the clay minerals and generating a higher hydrophobicity on the chalcopyrite particles.

Figure 5 shows the zeta potential of chalcopyrite, montmorillonite, and St-CTAB-VI NPs as a function of pH (Figure 5a). In the figure, it can be observed that the St-CTAB-VI NPs have a positive surface charge throughout the pH range studied (6–10), with a maximum value of 50 mV at pH 8 and a minimum value of 42 mV at pH 10, which is consistent with the report by Murga et al. (2022) [33]. In the case of the chalcopyrite, it had a negative potential in the range of pH studied, obtaining values between −6.06 mV (pH 6) and −40.69 mV (pH 10). Ma et al. (2016) [34] found the point of zero charge (PZC) of chalcopyrite to be at pH 5.7, a value that is consistent with the trend shown in this figure. On the other hand, in the case of montmorillonite, studies of the zeta potential of montmorillonite by Delgado et al. (1986) [35] and Au and Leong (2016) [36] reported an almost permanent negative charge of −25 to −35 mV in the pH range of 6 to 10. For montmorillonite in these tests, the zeta potential remained stable at a value of around −30 mV in the range of pH studied.

Figure 5b shows the interaction of chalcopyrite with montmorillonite and St-CTAB-VI NPs. In the figure it can be observed that the zeta potential of chalcopyrite obtained at the pH studied is like that at the values of zeta potential of montmorillonite, which indicates the occurrence of slime coating of this type of clay minerals on the chalcopyrite surface. On the other hand, when the St-CTAB-VI NPs interact with the chalcopyrite, the zeta potential of this mineral is changed to positive, obtaining a maximum value of 14.17 mV (pH 6) and minimum of 3.99 mV (pH 8), which indicates the adsorption of NPs on the chalcopyrite due to electrostactic forces. Also, at pH 8, the PZC was near zero, suggesting that the flotation should be near to optimal. It is important to note that montmorillonite and chalcopyrite presented negative charges at the pH studied, so electrostatics and van der Waals forces interaction can occur between this clay mineral and the nanoparticles. However, Yang et al., 2011 showed that nanoparticles functionalized with imidazole groups (VI) are designed to cause selective adsorption onto sulfide minerals, even in the presence of glass beads, which have a more negative charge than montmorillonite [32]. In this case, the interaction was due to electrostatic forces, considering that the nanoparticles had cationic charges and the presence of the imidazole group contributed a more stable deposition of nanoparticles onto the copper-rich surface In addition, considering the presence of NPs and montmorillonite in the system, NPs can be deposited faster on the surface of the chalcopyrite than the clay minerals, as the NPs have higher attractive forces and higher eletrophoretic mobility than montmorillonite [33].

### 3.2. Microflotation Test

Prior to the study of the effect of the interaction time of chalcopyrite with St-CTAB-VI NPs on the chalcopyrite recovery in the presence of montmorillonite, it was shown that the recovery of chalcopyrite decreases as the montmorillonite concentration increases in the presence of collector PAX (5.6 mg/g), obtaining a chalcopyrite recovery of 76.79% at pH 8 in the presence of 15% of this type of clay mineral. This decrease in chalcopyrite recovery due to the presence of montmorillonite has also been reported by researchers such as Farrokhpay and Ndlovu (2013) [6], who associated this behavior with the significant differences in the crystallinity, cation exchange capacity, and degree of swelling.

Figure 6a shows the results obtained for chalcopyrite recovery at different interaction times of the chalcopyrite with St-CTAB-VI NPs and mixtures of NPs (7.5 mg/g) and PAX (5.6 mg/g), in the presence of montmorillonite. In the figure, it can be observed that a higher chalcopyrite recovery was obtained for interaction times between 7 and 10 min and using mixtures of NPs and PAX.

Figure 6b shows a comparison of the best chalcopyrite recoveries obtained using PAX (5.6 mg/g), St-CTAB-VI (7.5 mg/g), and the mixtures of both reagents at the same concentrations. From the figure, it can be noted that the NPs alone obtained chalcopyrite recoveries similar to those reached with the conventional collector PAX. It was noted that when a mixture of NPs and collector PAX was used, the chalcopyrite recovery was improved, an increase of approximately 4% in the chalcopyrite recovery was obtained. However, it was noted that when a mixture of nanoparticles and collector PAX was used, the chalcopyrite recovery was improved. This could have been due to the fact that the incorporation of PAX (in the last 2 min of conditioning time) covered small hydrophilic spaces of molecular size available between the adhesion zone of nanoparticles on the chalcopyrite surfaces, reducing the probability of adhesion of clay minerals.

### 3.3. Turbidity Test

Figure 8a–d show the delta transmittance curve obtained as a function of time for the following suspensions prepared in 5 mM NaCl at pH 8: 15% montmorillonite (Figure 8a), chalcopyrite + 15% montmorillonite (Figure 8b), chalcopyrite + 15% montmorillonite + St-CTAB-VI (Figure 8c), and chalcopyrite + 15% montmorillonite + St-CTAB-VI + PAX (Figure 8d), using 7.5 mg of NPs and 5.6 mg of PAX per gram of chalcopyrite. The different graphics are divided into a lower zone of the tube between approximately 0 and 15 mm, a middle zone between approximately 16 and 65 mm, and a high zone between 65 and 80 mm, which is the upper part of the tube, as shown in Figure 7.

In the lower zone, all these figures show a zone of sedimentation, which occurs between 0 and 15 mm, depending on the conditions studied. In addition, to compare the middle zone, in the graphics obtained in the suspension with 15% montmorillonite and the mixture with chalcopyrite it can be observed that the suspension of 15% montmorillonite (Figure 8a) presented slower sedimentation than when chalcopyrite was added to the suspension (Figure 8b) due to the small distance between the different transmittance delta curves obtained as a function of time, indicating a higher rate of sedimentation of the suspension when the chalcopyrite is present. This provides evidence that the presence of chalcopyrite in the suspension increases the sedimentation rate of the montmorillonite due to the hetero-coagulation phenomenon that occurs between the clay minerals and the sulfide minerals. On the other hand, on the right-hand side of the graphics, corresponding to the high zone of the tube, it is possible to note that in the presence of chalcopyrite the transmittance delta lines as a function of time are wider than those obtained in the absence of this mineral. This behavior can be related to the presence of aggregates (chalcopyrite + montmorillonite).

Figure 8c shows the stability plots of the suspensions in the chalcopyrite + 15% montmorillonite + St-CTAB-VI. In this figure, it is possible to observe in the middle zone, between 16 and 65 mm, there is a higher sedimentation rate in the presence of nanoparticles due to the wide distance between the different transmittance delta curves obtained as a function of time compared to that obtained in the absence of nanoparticles (Figure 8b). Moreover, when a mixture of nanoparticles and PAX was used (Figure 8d), two were present in the medium zone, a clear zone between 16 and 28 mm, where the particles tend to settle, and a greater turbidity zone between 29 and 65 mm, which indicates that aggregates of suspended particles were maintained in this area. In addition, comparing the trends obtained in the upper zone, between 65 and 80 mm, it is possible to show that there are negative values of transmittance delta lines in both cases, which can be related to the presence of hydrophobic aggregates of chalcopyrite, montmorillonite, and nanoparticles in the suspension.

Figure 9 shows the sedimentation kinetics obtained for the suspensions above using 7.5 mg of NPs and 5.6 mg of PAX per gram of chalcopyrite. The blue curve corresponds to the suspension of 15% montmorillonite; the red curve, 15% montmorillonite + chalcopyrite; the cyan curve, 15% montmorillonite + chalcopyrite + St-CTAB-VI; and the pink curve, 15% montmorillonite + chalcopyrite + St-CTAB-VI + PAX.

To understand the phenomena, it is important to consider that these curves are divided into three zones. First, the sedimentation zone, corresponding to the linear part of each curve. Secondly, the transition zone corresponding to the concave zone, where the sedimentation velocity of the particles decreases because of the particles; and finally, the compaction zone, where the particles come into physical contact, causing their compression, since the physical interaction forces between the particles are especially intense. At the critical moment, the settled sludge has a concentration called critical; from that moment on, it begins to thicken until it reaches its fine concentration.

In the figure, the curve corresponding to the 15% montmorillonite suspension (blue curve) has a slow sedimentation, which is found between 0 and 6 h, a transition zone between 6 and 44 h, and after this time the compaction occurs. It is important to note that the maximum transmission value was 52%, indicating that turbidity is still present in the sample.

In contrast, the curve corresponding to the suspension of 15% montmorillonite + chalcopyrite (red curve) shows a faster sedimentation than that obtained in the montmorillonite suspension, with a sedimentation zone between 0 and 3 h. After this time, there is a transition zone between 3 h and 12 h, after this time, the compression of particles occurs. It is important to note that the maximum transmittance value reached was 63%, presenting a lower turbidity than that obtained in the absence of chalcopyrite. This behavior can be associated with the hetero-coagulation that occurs between both minerals, which causes a much faster sedimentation and lower turbidity due to the hetero-coagulation phenomena.

On the other hand, the curves corresponding to the suspension 15% montmorillonite + chalcopyrite + St-CTAB-VI (cyan curve) show a faster sedimentation than that obtained in montmorillonite + chalcopyrite (red curve), presenting a sedimentation zone between 0 and 2 h with only nanoparticles and between 0 and 1 h in the cases of mixtures (blue curve). After this time, there is a transition zone between 2 and 12 h (only NPs) and 1 and 12 h, in the cases of the mixtures of NPs and PAX, and after these times the compaction occurs. In addition, the maximum transmittance reached was from 54 and 52, respectively, presenting a turbidity near to that obtained with only montmorillonite. This behavior may be associated with the presence of different aggregates (chalcopyrite + nanoparticles + montmorillonite or just nanoparticles) that remain suspended in the suspension. Yang et al. 2013 [31] indicated that nanoparticle aggregation on an industrial scale may provide some positive impacts. For example, nanoparticle aggregates in sulfide mineral surfaces help to detach the slime and ease the bubble/attachment step during flotation.

## 4. Conclusions

The main conclusions from the current study are the following:The contact angle measurements and zeta potential results obtained for the chalcopyrite in the presence of montmorillonite provided evidence that the nanoparticle adsorption on the chalcopyrite surface contributed to increasing the hydrophobicity of the chalcopyrite in the presence of this type of clay mineral, reaching a contact angle value similar to that obtained with PAX. According to these findings it is possible to state that the slime coating phenomenon caused by the presence of this type of clay mineral will occur when nanoparticles are used in the process in a similar way as for the conventional collector (PAX).The mixture of St-CTAB-VI-02-50 and PAX contributed to increasing the contact angle of chalcopyrite in the presence of montmorillonite, which could be due to initial adsorption of the PAX collector contributing to a greater interaction of the NPs with the mineral surface and contributing to reducing the interaction with the clay minerals and generating a higher hydrophobicity on the chalcopyrite particles.Microflotation and turbidity test results showed that the nanoparticles could have the same behavior as the PAX collector when montmorillonite is present. However, the mixtures with both NPs and PAX contributed to promoting the generation of nanoparticle aggregates on sulfide mineral surfaces that helped to detach the slime and may facilitate the bubble/mineral attachment step during flotation.This study shows that the use of nanoparticles can contribute to achieving an improvement in the recovery of copper and the quality of concentrates obtained by flotation when low-grade minerals and high concentrations of clay minerals are present.Further studies are required to improve the performance of NPs in this type of system, evaluate the interactions with other types of minerals such as molybdenite and pyrite, and study their economic and environmental feasibility.

## Figures and Tables

**Figure 1 polymers-16-01682-f001:**
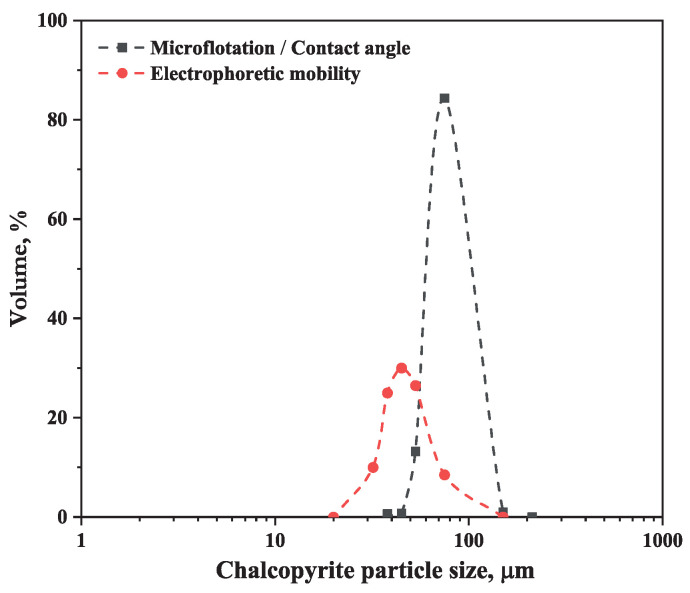
Particle size distribution of the chalcopyrite used for electrophoretic mobility, microflotation, and contact angle tests.

**Figure 2 polymers-16-01682-f002:**
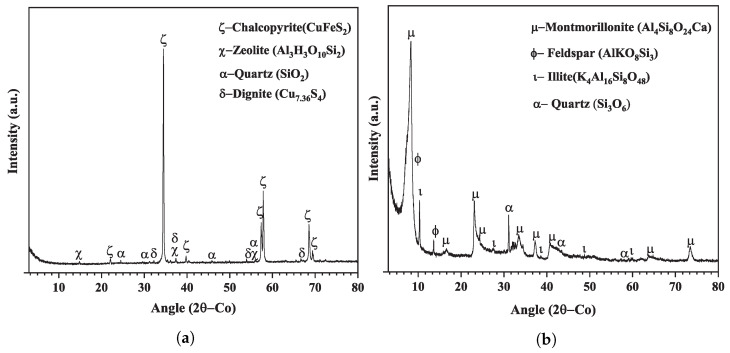
X-ray diffractogram for (**a**) chalcopyrite and (**b**) montmorillonite.

**Figure 3 polymers-16-01682-f003:**
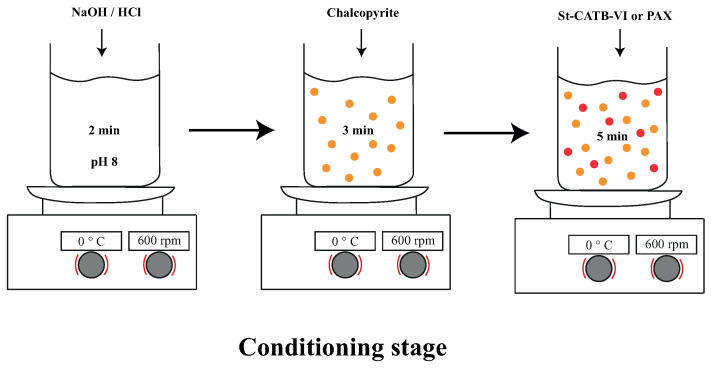
Conditioning time using NaCl solution at 5 mM, pH 8.0, and 600 rpm in the different mineral tests.

**Figure 4 polymers-16-01682-f004:**
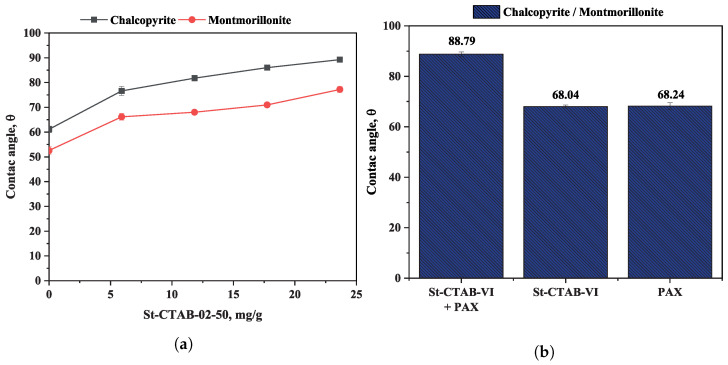
(**a**) Variation in contact angle for three different samples of chalcopyrite as a function of St-CTAB-VI concentration. (**b**) Comparison of contact angle using 5.6 mg/g of PAX and 7.5 mg/g of NPs for a chalcopyrite sample containing 15% clay.

**Figure 5 polymers-16-01682-f005:**
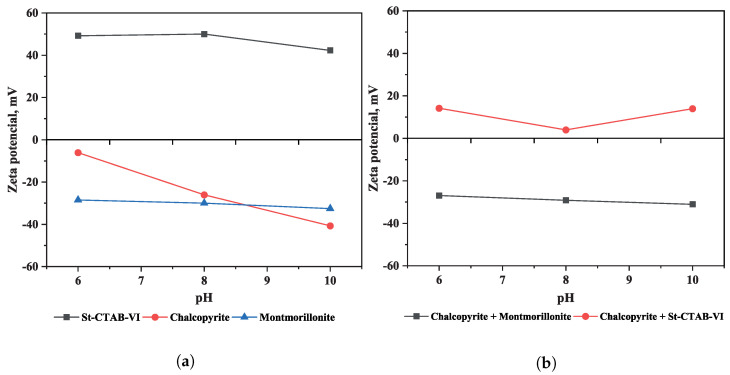
(**a**) Zeta potential as a function of pH of St-CTAB-VI nanoparticles, chalcopyrite, and montmorillonite. (**b**) Zeta potential of chalcopyrite as a function of pH in the presence of clay minerals (15%) and nanoparticles (7.5 mg/g).

**Figure 6 polymers-16-01682-f006:**
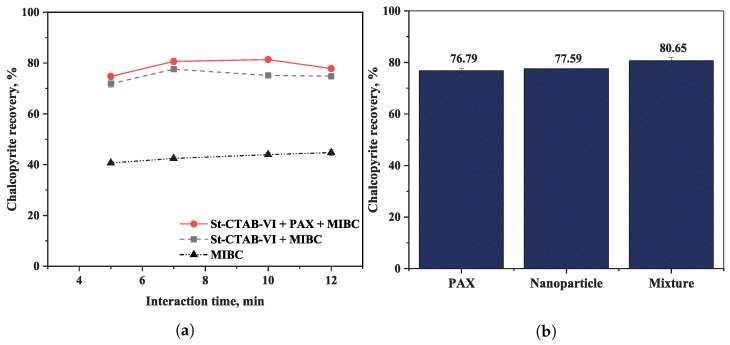
(**a**) Comparison of chalcopyrite recovery at different times of interaction of chalcopyrite with St-CTAB-VI-02-50 (7.5 mg/g) and mixtures with PAX (5.6 mg/g) in the presence of 15% montmorillonite and 15 ppm of MIBC. (**b**) Comparison of best chalcopyrite recoveries obtained using PAX, St-CTAB-VI, and mixtures of them.

**Figure 7 polymers-16-01682-f007:**
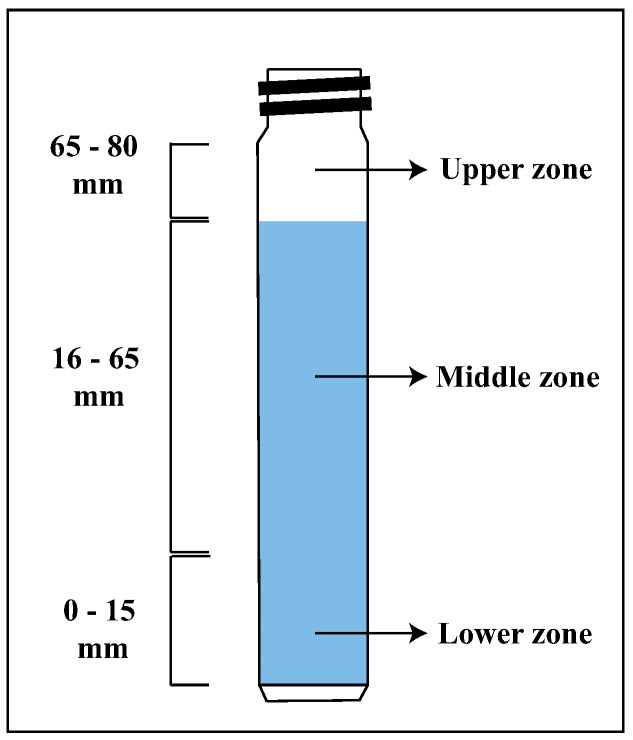
Turbidimeter cell.

**Figure 8 polymers-16-01682-f008:**
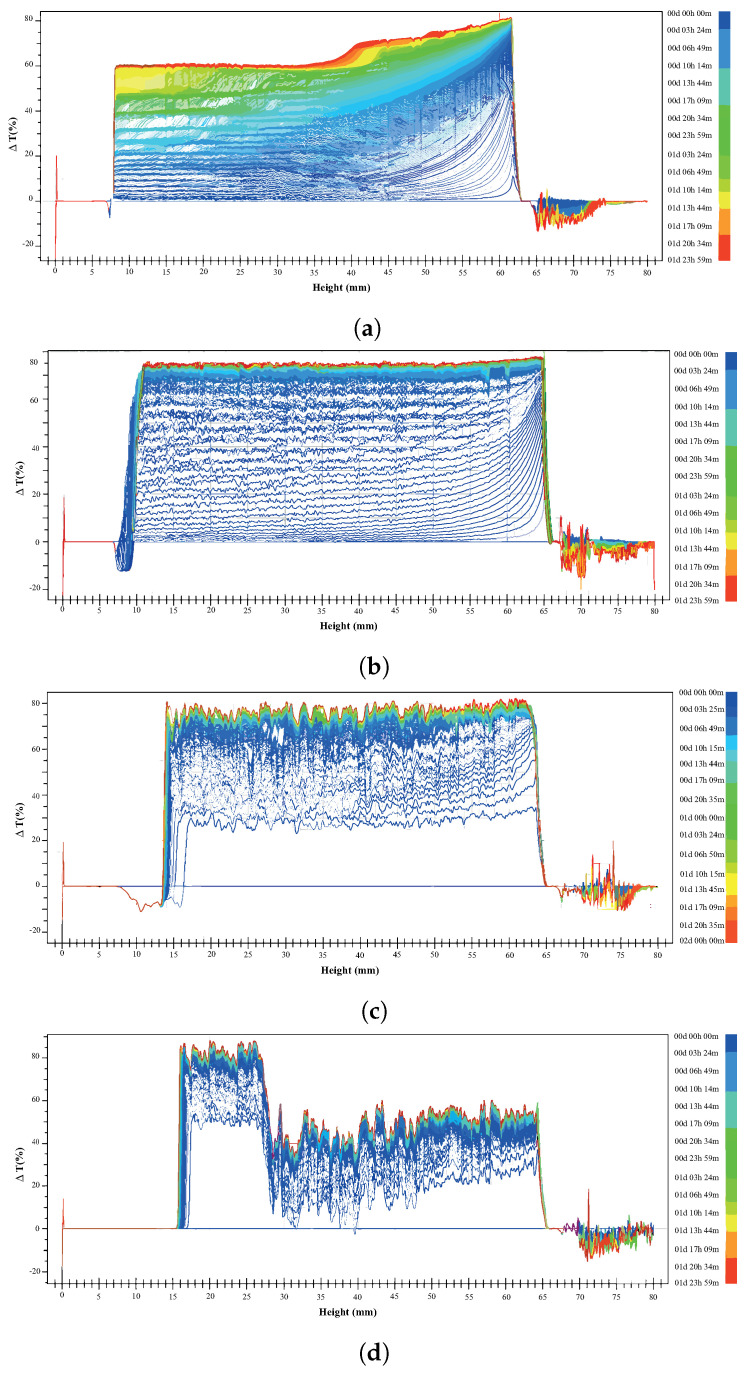
Delta transmittance as a function of time of suspensions of 15%montmorillonite (**a**), chalcopyrite + 15% montmorillonite (**b**), chalcopyrite + 15% montmorillonite + NPs (**c**), and chalcopyrite + 15% montmorillonite + NPs + PAX (**d**) using 7.5 mg of NPs and 5.6 mg of PAX per gram of chalcopyrite.

**Figure 9 polymers-16-01682-f009:**
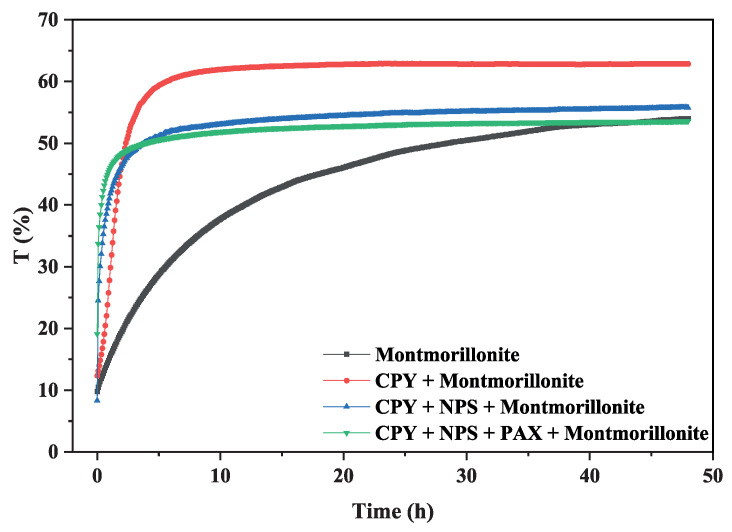
Chalcopyrite backscattering curve in the presence of montmorillonite and nanoparticles.

**Table 1 polymers-16-01682-t001:** Volume fraction data of chalcopyrite sample.

Test	150–75 μm, %	75–45 μm, %	45–32 μm, %
Microflotation/contact angle	85.37	13.98	0.64
Zeta potential	8.50	46.47	45.03

**Table 2 polymers-16-01682-t002:** Quantification of the crystalline phases in montmorillonite sample.

Chalcopyrite Sample			Montmorillonite Sample		
**Mineral Phase**	**ICSD or COD**	**Percentage [%]**	**Mineral Phase**	**ICSD or COD**	**Percentage [%]**
Chalcopyrite	ICSD-028894	93.0	Montmorillonite	COD-9002779	81.0
Zeolite	ICSD-082500	4.0	Quartz	COD-9012600	6.0
Quartz	ICSD-200722	1.0	Feldspar	ICSD-083532	10.0
Dignite	COD-9016668	2.0	Illite	COD-1010318	3.0

**Table 3 polymers-16-01682-t003:** St-CTAB-VI nanoparticle characterization at pH 8.0.

	Value
Hydrodynamic diameter, nm (PI *, %)	75.81 (20.89)
$D_{10}$; $D_{50}$; $D_{90}$, nm	28.98; 43.90; 81.04
Mobility, μm·cm/Vs	3.90
Zeta potential, mV	50.00 ± 1.26

* Polydispersity Index (%).

## Data Availability

Data are contained within the article.

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
