# Peer review of "On the Use of Styrene-Based Nanoparticles to Mitigate the Effect of Montmorillonite in Copper Sulfide Recovery by Flotation"

_polymers, 2024, doi:10.3390/polym16121682_

Round 1

Reviewer 1 Report

Comments and Suggestions for Authors

The paper titled " On the Use of Styrene-Based Nanoparticles to Mitigate the Effect 4 of Montmorillonite in the Copper Sulfide Recovery by Flotation” This study is of interest and a certain significance to the field of minerals processing. But some modifications need before accepted for publication.

1-     The study compares contact angle measurements among different conditions and treatments, but the significance of these comparisons is not fully explained. A more comprehensive discussion on the implications of these findings would strengthen the analysis.

2-     The manuscript contains grammatical errors and unclear phrasing in several passages, affecting the clarity and coherence of the text. For example, the statement "The results was analyzed using various statistical methods" lacks clarity regarding the specific statistical methods employed. Additionally, the conclusion section's statement "The findings show a significant impact on the experimental outcomes" is vague and could be clarified for better understanding. Addressing these issues would improve the manuscript's readability and overall quality.

3-     The study discusses the decline in chalcopyrite recovery with increasing montmorillonite concentration, which aligns with prior research findings. However, the explanation for this trend lacks depth, requiring further elucidation of the underlying mechanisms.

4-     The interpretation of the observed increase in chalcopyrite recovery with the use of a mixture of nanoparticles and PAX lacks specificity. Further clarification is needed to explain how the incorporation of PAX contributes to improved chalcopyrite recovery, particularly in terms of its interaction with nanoparticles and clay minerals.

5-     The presentation of figures 6a-6d lacks clarity, making it difficult for readers to discern specific trends and phenomena. Improved labeling and graphical representation would enhance the readability and interpretation of the data.

6-     The captions of Tables and Figures shall be self-explanatory (i.e., should include operating conditions).

7-      The error bars should be provided.

Comments on the Quality of English Language

Minor editing of English language required

Author Response

Please refer to the answer to the reviewer 1:

  1. The study compares contact angle measurements among different conditions and treatments, but the significance of these comparisons is not fully explained. A more comprehensive discussion on the implications of these findings would strengthen the analysis.

R/Thank you for your comments. The paragraph was improved according to this comment. Please refer to the updated version of the manuscript.

  1. The manuscript contains grammatical errors and unclear phrasing in several passages, affecting the clarity and coherence of the text. For example, the statement "The results was analyzed using various statistical methods" lacks clarity regarding the specific statistical methods employed. Additionally, the conclusion section's statement "The findings show a significant impact on the experimental outcomes" is vague and could be clarified for better understanding. Addressing these issues would improve the manuscript's readability and overall quality.

R/: The manuscript was revised but we could not find the sentences highlighted by the reviewer.

  1. The study discusses the decline in chalcopyrite recovery with increasing montmorillonite concentration, which aligns with prior research findings. However, the explanation for this trend lacks depth, requiring further elucidation of the underlying mechanisms.

R/ The manuscript was revised but we could not find the sentences highlighted by the reviewer.

There are no flotation tests associated to the chalcopyrite recovery as a function of the montmorillonite concentration.

  1. The interpretation of the observed increase in chalcopyrite recovery with the use of a mixture of nanoparticles and PAX lacks specificity. Further clarification is needed to explain how the incorporation of PAX contributes to improved chalcopyrite recovery, particularly in terms of its interaction with nanoparticles and clay minerals.

Thank you for your comments. The paragraph was improved according to this comment. Please refer to the updated version of the manuscript

  1. The presentation of figures 6a-6d lacks clarity, making it difficult for readers to discern specific trends and phenomena. Improved labeling and graphical representation would enhance the readability and interpretation of the data

R/Thank you for your comments. The figures were improved according to this comment and the structure of this section was improved. Please refer to the updated version of the manuscript

  1. The captions of Tables and Figures shall be self-explanatory (i.e., should include operating conditions).

R/Thank you for your comments. The caption in Figure 6 and 7 were improved according to this comment and the structure of this section was improved. Please refer to the updated version of the manuscript.

  1. The error bars should be provided

R/ The error bars are added in the different curves. However, they cannot be seen in the graph because they are very small.

Reviewer 2 Report

Comments and Suggestions for Authors

 The manuscript under consideration is devoted to the search for solutions to the problem of reducing chalcopyrite losses during its flotation enrichment by using a polymer additive. The topic of the work is very relevant and corresponds to the direction of the magazine.

In short, the problem being studied is as follows. The initial mineral raw materials extracted from not very valuable copper deposits, in addition to chalcopyrite itself - copper sulfide - contains a significant amount of a clay component - montmorillonite (MMT). After extraction, the mineral rock is crushed, while microparticles of chalcopyrite and MMT are formed, which differ in their ability to be wetted with water, namely, chalcopyrite particles are hydrophobic, and MMT particles are hydrophilic. This difference is used to separate chalcopyrite particles from MMT by flotation. Usually, a surfactant (potassium amylxanthogenate) is added to an aqueous dispersion containing crushed particles of mineral raw materials. The dispersion is blown by an air stream. In this case, a foam is formed that selectively extracts hydrophobic chalcopyrite particles, leaving hydrophilic montmorillonite particles in solution. However, the efficiency of such technology is not very high, since MMT particles interact with the surface of chalcopyrite particles, reducing their hydrophobicity. To overcome this effect, the authors of the manuscript proposed adding positively charged polystyrene (PS) nanoparticles obtained by emulsion polymerization of styrene in the presence of a cationic surfactant to the dispersed system. In the work, the values of the positive zeta potential of PS and chalcopyrite particles, which turned out to be negative, were measured by electrophoresis. This suggests the possibility of electrostatic interaction of PS and chalcopyrite particles. Indeed, this interaction has been confirmed experimentally. It leads to an increase in the hydrophobicity of the surface of montmorillonite, which, in turn, increases the yield of chalcopyrite during its flotation enrichment. The best result was obtained by using polystyrene nanoparticles in combination with a traditionally used flotation agent - potassium amylxanogenate. The advantage of the manuscript should include an ingenious technique for determining the angle of wetting the chalcopyrite surface with water using plates pressed from finely dispersed chalcopyrite powder. In addition, the manuscript uses modern research methods such as dynamic light scattering, measurement of precipitation kinetics by turbidity measurement, etc.

In general, it can be concluded that the work was performed at a good scientific and methodological level, it obtained a significant beneficial effect. Manuscript and can be recommended for publication in the journal Polymers with minimal corrections.

Note on the manuscript: it is necessary to edit the text corresponding to the mark 260.

The following remarks should be added:

1) Figure 5 shows the yield curves of chalcopyrite and + 15% montmorillonite in the presence of PS and PS+PAX additives, For comparison, similar data for a system without additives should be provided on the same graph.

2) In Fig.8, the experiment for a mixture of chalcopyrite + 15% montmorillonite in the presence of PS+AX additives corresponds to a blue curve, and in the description the same curve is called pink (label 305, 331).

3) Label 259-265 "...However, an increase of approximately 4% in the chalcopyrite recovery was obtained to use the mixture PAX and NPs. These results suggest that nanoparticles. However, it was noted that 258-265

when a mixture of nanoparticles and collector PAX was used, the chalcopyrite recovery was improved.

4) Label 353 “... sizes which a higher hydrophobicity on the chalcopyrite particles...” to edit text

5)  Label"358-362 “...to detach the slime and ease the may facilitate the bubble/mineral...” to edit text.

Comments on the Quality of English Language

Minor corrections

Author Response

Dear Reviewer, please refer to the answers to your comments below:

  • Figure 5 shows the yield curves of chalcopyrite and + 15% montmorillonite in the presence of PS and PS+PAX additives, For comparison, similar data for a system without additives should be provided on the same graph.

R/: This comment was considered. Please refer to the updated version of the manuscript.

  • In Fig.8, the experiment for a mixture of chalcopyrite + 15% montmorillonite in the presence of PS+PAX additives corresponds to a blue curve, and in the description the same curve is called pink (label 305, 331).

R/: This comment was considered. Please refer to the updated version of the manuscript.

  • Label 259-265 "...However, an increase of approximately 4% in the chalcopyrite recovery was obtained to use the mixture PAX and NPs. These results suggest that nanoparticles. However, it was noted that 258-265 when a mixture of nanoparticles and collector PAX was used, the chalcopyrite recovery was improved.

R/: This comment was considered. Please refer to the updated version of the manuscript

  • Label 353 “... sizes which a higher hydrophobicity on the chalcopyrite particles...” to edit text

R/: This comment was considered. Please refer to the updated version of the manuscript.

  • Label"358-362 “...to detach the slime and ease the may facilitate the bubble/mineral...” to edit text.

R/: This comment was considered. Please refer to the updated version of the manuscript.

Reviewer 3 Report

Comments and Suggestions for Authors

The submitted manuscript currently deals with a key issue, namely the extraction of raw materials (in this case, copper) from natural deposits that are made up not only of these raw materials but also of other materials, mostly clay minerals.

This work is devoted in detail to the simulation of the flotation process (test) after the turbidity test. However, because of its complexity, it is necessary to elaborate the material characteristics of the experimental samples studied in detail. I have the following comments on the requirement:

*/ The particle size of the chalcopyrite sample is ambiguously determined. It is not possible to determine with only one value for heterogeneous materials. Complete the work with a distribution representation and then evaluate it in more detail.

*/ from Chapter 2.1 I conclude that measurements of the Zeta potential, wetting angle, and microflotation test were carried out on different size fractions of experimental samples. However, this is unacceptable. These characteristics are dependent on the particle size. It is necessary to add measurements for individual size fractions.

*/ The hydrodynamic diameter of prepared nanoparticles had a heterogeneous distribution. This is very key to the flocculation process ... was it heterogeneity in the form of a broad distribution curve or were there multiple size fractions in the volume?

*/ Explain the importance of the mobility of the prepared nanoparticles in relation to the microflotation test.

*/ To clarify "the slime-coating phenomena caused by the presence of different types of clay mineral", it is very important to know how the materials interact with each other. It is desirable that these materials be evaluated using XRD analysis (to analyse the effect of the presence of chalcopyrite/montmorillonite versus NPs and mixtures of NPs+PAX.)

*/ from which characteristic do you conclude that the nanoparticles were adsorption on the chalcopyrite surface?

*/ In the conclusion, I lack an evaluation of which of the clay minerals used is more suitable (more ecological and economical) for obtaining Cu raw materials.

The manuscript requir a major revision.

Author Response

Dear Reviewer

Please refer to the answers to your comments below. Thank you:

  1. The particle size of the chalcopyrite sample is ambiguously determined. It is not possible to determine with only one value for heterogeneous materials. Complete the work with a distribution representation and then evaluate it in more detail.

R/ This comment was considered. Please refer to the updated version of the manuscript.

  1. from Chapter1 I conclude that measurements of the Zeta potential,wetting angle, and microflotation test were carried out on different size fractions of experimental samples. However, this is unacceptable. These characteristics are dependent on the particle size. It is necessary to add measurements for individual size fractions.

R/: You are right, the size fraction used in micro-flotation tests and wetting angle measurements were the same. However, in the case of zeta potential measurements, the size used for these measures was less to 75 µm due that the size range of electrophoretic mobility test in litesizer 500 is between 38 nm and 100 µm.

  1. Thehydrodynamic diameter of prepared nanoparticles hada heterogeneous distribution. This is very key to the flocculation process ... was it heterogeneity in the form of a broad distribution curve or were there multiple size fractions in the volume?

R/ This comment was considered. Please refer to the updated version of the manuscript.

  1. Explain the importance of the mobility of the prepared nanoparticles in relation to the microflotation test

R/ This comment was considered. Please refer to the updated version of the manuscript.

  1. Toclarify "the slime-coating phenomena caused by the presence of different types of clay mineral", it is very important to know how the materials interact with each other. It is desirable that these materials be evaluated using XRD analysis (to analyse the effect of the presence of chalcopyrite/montmorillonite versus NPs and mixtures of NPs+PAX.)

R/Thank you for your comments. The slime-coating phenomena was explain with more detail and also the effect of the presence of chalcopyrite/montmorillonite versus NPs and mixtures of NPs+PAX . Please refer to the updated version of the manuscript.

In adittion, due to the sample size used in the microflotation tests, it was not possible to perform this XRD analysis. However, subsequent studies carried out in laboratory cells that operate with a larger amount of sample will consider this analysis.

  1. / from which characteristic do you conclude that the nanoparticles were adsorption on the chalcopyrite surface?

R/ The adsorption of NPs on the chalcopyrite surface is due to electrostactic forces. Also, at pH 8, the pz was near to zero, indicating that flotation should be near to the optimal of flotation process.

Also, a first adsorption of PAX collector may be contributing to a greater interaction of the NPs with the mineral surface and contribute to reduce the interaction with the clay minerals and generate a higher hydrophobicity on the chalcopyrite particles.

  1. In the conclusion, I lack an evaluation of which of the clay minerals used is more suitable (more ecological and economical) for obtaining Cu raw materials.

R/ According to this comment new conclusions were made. Please refer to the updated version of the manuscript

Round 2

Reviewer 1 Report

Comments and Suggestions for Authors

The authors have carefully addressed all the issues I raised previously. I recommend it for publication 

Reviewer 3 Report

Comments and Suggestions for Authors

The manuscript has been revised accordingly and the amended version can be accepted.